# Aqueous Fraction from *Cucumis sativus* Aerial Parts Attenuates Angiotensin II-Induced Endothelial Dysfunction In Vivo by Activating Akt

**DOI:** 10.3390/nu15214680

**Published:** 2023-11-04

**Authors:** Celeste Trejo-Moreno, Zimri Aziel Alvarado-Ojeda, Marisol Méndez-Martínez, Mario Ernesto Cruz-Muñoz, Gabriela Castro-Martínez, Gerardo Arrellín-Rosas, Alejandro Zamilpa, Jesús Enrique Jimenez-Ferrer, Juan Carlos Baez Reyes, Gladis Fragoso, Gabriela Rosas Salgado

**Affiliations:** 1Facultad de Medicina, Universidad Autónoma del Estado de Morelos, Cuernavaca 62350, Morelos, Mexico; trejomc@hotmail.com (C.T.-M.); zimrihazi@gmail.com (Z.A.A.-O.); mario.cruz@uaem.mx (M.E.C.-M.); garrellin@up.edu.mx (G.A.-R.); 2Departamento de Sistemas Biológicos, División de Ciencias Biológicas y de la Salud, Universidad Autónoma Metropolitana-Xochimilco, Ciudad de México 04960, Mexico; mm.mary87@gmail.com; 3Doctorado en Ciencias Biológicas y de la Salud, Universidad Autónoma Metropolitana, Ciudad de México 04960, Mexico; gcm_19@hotmail.com; 4Facultad de Ciencias de la Salud, Universidad Panamericana, Ciudad de México 03920, Mexico; 5Centro de Investigación Biomédica del Sur, Instituto Mexicano del Seguro Social, Xochitepec 62790, Morelos, Mexico; azamilpa_2000@yahoo.com.mx (A.Z.); enriqueferrer_mx@yahoo.com (J.E.J.-F.); 6Escuela Nacional Preparatoria No. 1, Universidad Nacional Autónoma de México, Ciudad de México 16030, Mexico; baezrjc@yahoo.com.mx; 7Instituto de Investigaciones Biomédicas, Universidad Nacional Autónoma de México, Ciudad de México 04510, Mexico

**Keywords:** *Cucumis sativus*, endothelial dysfunction, hypertension, pAkt, angiotensin-II, mice

## Abstract

Background: Endothelial dysfunction (ED) is a marker of vascular damage and a precursor of cardiovascular diseases such as hypertension, which involve inflammation and organ damage. Nitric oxide (NO), produced by eNOS, which is induced by pAKT, plays a crucial role in the function of a healthy endothelium. Methods: A combination of subfractions SF1 and SF3 (C4) of the aqueous fraction from *Cucumis sativus* (Cs-Aq) was evaluated to control endothelial dysfunction in vivo and on HMEC-1 cells to assess the involvement of pAkt in vitro. C57BL/6J mice were injected daily with angiotensin II (Ang-II) for 10 weeks. Once hypertension was established, either Cs-AqC4 or losartan was orally administered along with Ang-II for a further 10 weeks. Blood pressure (BP) was measured at weeks 0, 5, 10, 15, and 20. In addition, serum creatinine, inflammatory status (in the kidney), tissue damage, and vascular remodeling (in the liver and aorta) were evaluated. Cs-AqC4 was also tested in vitro on HMEC-1 cells stimulated by Ang-II to assess the involvement of Akt phosphorylation. Results: Cs-AqC4 decreased systolic and diastolic BP, reversed vascular remodeling, decreased IL-1β and TGF-β, increased IL-10, and decreased kidney and liver damage. In HMEC-1 cells, AKT phosphorylation and NO production were increased. Conclusions: Cs-AqC4 controlled inflammation and vascular remodeling, alleviating hypertension; it also improved tissue damage associated with ED, probably via Akt activation.

## 1. Introduction

Endothelial dysfunction (ED) is a marker of vascular damage and a precursor of cardiovascular diseases such as hypertension and renal failure, which are leading causes of death in the world [1]. There are many factors that affect endothelial cell function, such as macrophage phenotype (according to the inflammatory process), smooth muscle cell phenotype [2], reactive oxygen species (ROS), oxLDL, advanced glycosylation end products, and particularly for this work, angiotensin II (Ang-II) [3]. The latter is the main effector molecule of the renin–angiotensin system. It activates several signaling pathways that regulate vascular biology, and when altered, it is a major contributor to the development of vascular diseases such as ED [4]. ED is characterized by oxidative stress, inflammation, vascular remodeling, vasoconstriction, and a prothrombotic state [5].

The pathological basis of ED is an imbalance in the synthesis, release, or effects of factors synthesized by endothelial cells [3], particularly nitric oxide (NO), a key regulator of the cardiovascular system [6] whose deficit impairs the ability of the endothelium to modulate the physiological functions of the vascular bed [3]. The main source of NO production in the endothelium is the endothelial variant of the enzyme nitric oxide synthase (eNOS), whose activation depends on protein kinase B (Akt) [7].

In ED, the Akt/eNOS/NO signaling pathway is altered either by the effect of Ang-II, reactive oxygen species (ROS), or inflammatory mediators (IL-6, TNF-α, and protein C-reactive), among other stimuli [8,9]. The failure of NO to diffuse to the vascular smooth muscle cells (VSMCs) has two main effects: (1) actomyosin relaxation is prevented, thus maintaining a vasoconstricted state and (2) VSMC proliferation is induced, causing vessel walls to thicken (remodel) [9,10].

On the other hand, stimuli that promote ED, including ROS, Ang-II, oxLDL, and advanced glycosylation end products, also activate NF-κB, which induces the synthesis of proinflammatory cytokines (IL-1b, IL-6, TNF-α), chemokines (MCP-1), and adhesion molecules (VCAM-1, ICAM-1, and E-selectin). This proinflammatory environment promotes the adhesion, transmigration, and activation of inflammatory cells (neutrophils, monocytes, and T lymphocytes), leading to an overexpression of cytokines, chemokines, growth factors, and ROS, which further contribute to inflammation and damage [4] to target organs such as kidney and liver [11].

Previous studies have reported that *Cucumis sativus* has antioxidant and anti-inflammatory activities [12,13,14]. In our research group, we observed that the subfractions SF1 and SF3 of the aqueous fraction of *C. sativus* aerial parts showed protective effects in vitro on Ang-II-induced ED using HMEC-1 cells. The main components found in both subfractions were glycine, arginine, asparagine, lysine, and aspartic acid [15]. This study aims to assess in vivo the ability of a combination of SF1 and SF3, labeled as C4, to control hypertension, vascular remodeling, inflammation, and renal and hepatic damage induced by Ang-II administration, as well as to preliminarily elucidate the mechanism of action underlying its activity against ED in vitro.

## 2. Material and Methods

### 2.1. Plant Material and Fractions Preparation

Aerial parts of *C. sativus*, including stems, leaves, and fruits, were collected from a commercial, pesticide- and fertilizer-free crop field in Xochitepec, Morelos, Mexico, from July to August. Margarita Avilés Flores and Macrina Fuentes Mata (Herbario del Jardín Etnobotánico, INAH, Cuernavaca, Mexico) identified all collected specimens. Voucher specimens were kept for future reference (INAH-Morelos 3001), protected from light, and dried at room temperature (RT).

The dry material was milled in a Pulvex electric mill (Büchi Labortechnik, Flawil, Switzerland) to particles < 4 mm in diameter. Maceration with a 60:40 (*v*/*v*) ethanol/water mixture at RT yielded a hydroalcoholic extract. This extract was distilled under reduced pressure and lyophilized. Fifty grams of this extract were bi-partitioned with ethyl acetate/water to obtain the aqueous fraction. The aqueous fraction was then distilled under reduced pressure, suspended in 700 mL of methanol for 24 h, filtered, concentrated at a rotary-evaporator (Laborota 4000, Heidolph, Schwabach, Germany), and freeze-dried to produce the subfraction SF1. The organic layer was suspended in 700 mL of acetone, yielding the subfractions SF2 (soluble) and SF3 (precipitate), which finally were concentrated and freeze-dried.

### 2.2. Cell Culture

Human microvascular endothelial cells-1 (HMEC-1) were kindly donated by Aida Castillo, Department of Physiology, CINVESTAV-IPN, Mexico City. The cells were cultured in MCDB-131 basal medium plus 10% fetal serum bovine (FBS), L-glutamine 10 mM, penicillin–streptomycin 100 U/mL (Invitrogen, Carlsbad, CA, USA), endothelial growth factor 10 ng/mL, and hydrocortisone 1 μg/mL (Sigma-Aldrich, St. Louis, MO, USA) at 37 °C under 5% CO_2_. Cell passages 3–8 were used in all experiments. The cells were cultured in medium plus Ang-II 5000 nM and either 10 μmol losartan or C4 (10 μg/mL of SF1/SF3) for 24 h.

### 2.3. Western Blot

PBS-washed HMEC-1 cells were incubated with lysis buffer (20 mM Tris pH 7.4, 150 mM NaCl, 1 mM EDTA pH 7.4, 0.5% Triton X-100, 0.1% SDS, and 0.5% sodium deoxycholate) and a cocktail of phosphatase and protease inhibitors (Roche, Basel, Switzerland) for 15 min. Lysates were centrifuged at 9000× *g* for 10 min at 4 °C. Proteins in supernatants were quantified using the Bradford method. A supernatant volume equivalent to 20 μg of protein was boiled for 5 min in sample buffer, separated by SDS-PAGE on 10% acrylamide gels, and electro-transferred to a PVDF membrane (Merck, Darmstadt, Germany) using transfer buffer (25 mM Tris pH 8.5, 193 mM glycine, and 20% methanol). The membranes were blocked for 1 h in blocking buffer (Tris-buffered saline and 0.1% Tween-20 [TBS-T]) plus 2% bovine serum albumin (BSA). Then, the membranes were incubated for 2 h with the primary antibody: 1:1000 anti-Akt, 1:1000 anti-phospho-Akt, or anti-β-actin (Cell Signaling Technology, Danvers, MA, USA). After washing five times for 5 min with TBS-T, the blots were incubated with 1:2000 rabbit anti-mouse antibody (Cell Signaling, Danvers, MA, USA) for 2 h. Binding was detected with the SuperSignal West Dura Extended Duration Substrate solution (Thermo Scientific, Waltham, MA, USA). The bands were analyzed with the ImageJ software v.1.54 (National Institute of Health, Bethesda, MD, USA), using anti-actin for normalization.

### 2.4. Animals

Male C57BL/6J mice (aged 8–10 weeks) were retrieved from our animal facility. The guidelines of the National Institutes of Health Guide for the Care and Use of Laboratory Animals were followed in all experiments, and experimental protocols were reviewed and approved by the FM-UAEM Ethical Committee for the Care and Use of Laboratory Animals (Permit No. 008/2016). The mice were housed in four groups of five animals each and kept in the animal house of the Faculty of Medicine, Universidad Autónoma del Estado de Morelos, under pathogen-free conditions, constant temperature (21–23 °C) and humidity (45–50%), with a 12 h light/dark cycle. All experiments were repeated three times. ED was established within the first 10 weeks of the experiment. Once the blood pressure (BP) of the mice increased by 15% or more from baseline values (systolic or diastolic), the animals were considered hypertensive. All treatments were administered during the next 10 weeks of the experiment. The control group was given saline solution only, and the other three groups were treated daily with Ang-II (0.1 μg/kg) via intraperitoneal (i.p.) injection throughout the experiment [11]. One treated group was left as control, receiving only Ang-II; another was orally treated with Losartan (10 mg/kg/day); and the last group was orally administered with C4 (19 mg/kg of SF1 and 6.7 mg/kg of SF3) diluted in water. To establish the amount of each fraction to be administered, a pilot test was conducted. In this pre-test, the effect of 50 mg/kg of the hydroalcoholic extract or ethyl-acetic/aqueous fractions was evaluated in a murine model of ED, taking into account the yield of SF1 and SF3 per gram of aqueous fraction. At the end of the experiment, the mice were killed by CO_2_ inhalation and exsanguinated.

### 2.5. Blood Pressure Measurement

Systolic blood pressure (SBP) and diastolic blood pressure (DBP) were measured in anesthetized mice (Xylazine, 10 mg/kg, i.p.) using a noninvasive blood pressure system (LE5002 system, Panlab, Barcelona, Spain) by the tail-cuff method. BP was reported as the average of seven successive measurements.

### 2.6. Organ Retrieving

Upon the last BP measurement, mice were perfused with cold PBS under anesthesia (sodium pentobarbital, 30 mg/kg, i.p.). Then, blood, liver, kidneys, and aorta were obtained.

### 2.7. Biochemical Analysis of Plasma and Liver

Plasma samples were obtained by centrifugation of heparinized whole blood at 1000× *g* for 10 min and stored at −80 °C for further analysis. Creatinine, urea, and uric acid levels were measured with commercial assay kits (Química Clínica Aplicada, S.A., Amposta, Tarragona, Spain), following the manufacturer’s instructions. Livers were macerated with ice-cold 1:10 PBS (*w*/*v*). The suspensions were centrifuged at 1000× *g* for 8 min at 4 °C, and the supernatants were used to quantify triglycerides and glucose using commercial assay kits (Química Clínica Aplicada, S.A.).

### 2.8. ELISA

The kidneys were stored at −80 °C until processed. All organs were macerated in ice-cold PBS-0.1% PMSF 1:5 *w*/*v*. The resulting suspensions were centrifuged, and supernatants were frozen at −80 °C until processed. Cytokines were quantified with commercial ELISA kits, following the manufacturer’s instructions. OptEIA Mouse IL-1β, IL-6, TNF-α, IL-10 (BD, Franklin Lakes, NJ, USA), and mouse TGF-β (Applied Biosystems, Foster City, CA, USA) ELISA kits were used. Briefly, 96-well flat-bottomed ELISA plates were coated with the respective capture antibody and incubated overnight at 4 °C. Non-specific binding was blocked by incubating for 1 h at RT with PBS-1% BSA. Aqueous kidney extracts were added and incubated for 2 h at RT. Then, the plates were incubated with the respective anti-cytokine-HRP antibody for 1 h at RT. Tetramethylbenzidine was added after 30 min of incubation in the dark, and the reaction was stopped with 2N H_2_SO_4_. Absorbance at 450 nm was measured at 37 °C in a VERSAmax ELISA plate reader (Molecular Devices, San Jose, CA, USA). Standard curves for each cytokine were prepared, and cytokine concentrations were reported as pg/mg protein.

### 2.9. Histopathology

Kidneys, aorta, and liver were fixed in buffered formalin (PBS, 10% formaldehyde pH 7.0). Then, the tissues were dehydrated and embedded in paraffin. Tissue sections (5 μm) were transferred to poly-l-lysine-coated slides (Sigma), deparaffinized, and rehydrated. For histopathological studies, the slides were stained with either the hematoxylin-eosin or Masson trichrome stain. All slides were observed under an ECLIPSE 80i microscope (Nikon, Tokyo, Japan) and analyzed using the software Metamorph v.6.1. (Molecular Devices, San Jose, CA, USA).

### 2.10. Statistical Analysis

Results are reported as mean ± standard error (SE). Data were analyzed using the software InStat v.3.05 (GraphPad, San Diego, CA, USA). Differences between treatments were assessed by ANOVA and Tukey’s post hoc test; they were regarded as significant for *p* < 0.05.

## 3. Results

### 3.1. C4 Regulates Blood Pressure

Increased BP is one of the most evident consequences of ED [5]. The kinetics of systolic (SBP, Figure 1A) and diastolic BP (DBP, Figure 1B) in our mice are shown in Figure 1. After 10 weeks of Ang-II administration, SBP increased (*p* < 0.05) by 22.2% (141.5 ± 3.4 mmHg), whilst DBP increased by 29.9% (78.4 ± 2.9 mmHg) with respect to the control group (SBP, 117.3 ± 2.9 mmHg; DBP 60.3 ± 3.3 mmHg) (*p* < 0.05). After five weeks of treatment with Losartan, both pressure values decreased significantly (SBP, 127.1 ± 2.8 mmHg; DBP 71.5 ± 2.1 mmHg) (*p* < 0.05) with respect to the values in week 10, restoring BP to the level of the control group. On the other hand, both pressure values gradually decreased in mice receiving C4 (SBP, 136.9 ± 2.5 mmHg; DBP, 75.0 ± 1.9 mmHg). Although the animals still showed significantly higher BP values with respect to the control group after 5 weeks of treatment (9.2% and 13.5% higher in SBP and DBP, respectively), C4 treatment decreased both blood pressure values to levels comparable to the control group after 10 weeks of treatment. In contrast, the Ang-II group showed BP values 19.9% (SBP, 145.7 ± 4.0 mmHg) and 27.8% (DBP 83.5 ± 3.2 mmHg) higher than those in the control group (SBP, 121.6 ± 2.3 mmHg; DBP 66.2 ± 2.0 mmHg).

### 3.2. C4 Stimulates Akt Phosphorylation and NO Production

To determine whether the effect of C4 on blood pressure was linked to both increased NO bioavailability and Akt phosphorylation status, these parameters were analyzed in HMEC-1 endothelial cells cultured in the presence of Ang-II, plus either C4 or Losartan for 24 h. On one hand, treatment with Ang-II decreased NO production by 22.4% (188.9 ± 10.1 µM; *p* < 0.05), whereas treatment with Losartan (239.1 ± 11.5 µM) or C4 (233.9 ± 8.5 µM) prevented this decrease (Figure 2A) despite the continued presence of Ang-II. No changes in this parameter were observed in control cells cultured with C4 (246.4 ± 10.3 µM) or Losartan (232.4 ± 12.3 µM). NO is produced from arginine by eNOS, an enzyme that is activated by Akt, a serine-threonine kinase [16]. To determine the mechanism by which C4 prevents the decrease in NO levels, phosphorylated Akt expression was measured. Our results indicate that basal levels of total Akt remain unchanged regardless of treatment (Figure 2B–C). However, Losartan increased Akt phosphorylation by 71% (3.6 ± 0.4 AU; *p* < 0.05), whereas C4 increased it even more (156%, 5.4 ± 0.7 AU; *p* < 0.05) (Figure 2B,D).

### 3.3. C4 Reverses Vascular Remodeling

A key effect of reduced NO bioavailability is the proliferation of VSMCs, which increases the thickness of blood vessel walls, decreasing their lumen [17] and contributing to hypertension. Our results show that Ang-II promoted the thickening of the tunica media of both small (hepatic arteriole) (Figure 3B,M) (156%; 21.3 ± 6.7 AU) (*p* < 0.05) and large arteries (aorta) (Figure 3F,J,N) (200%; 0.6 ± 0.1 AU) (*p* < 0.05) with respect to the control group (8.3 ± 4.0 and 0.2 ± 0.04 AU, respectively) (Figure 3A,E,I,M,N). Both losartan (47.4%, 11.2 ± 4.0 and 53.2%, 0.30 ± 0.02 AU, respectively) (Figure 3C,G,K,M,N) and C4 (43.9%, 14.3 ± 2.5 and 59.4%, 0.26 ± 0.03 AU, respectively) decreased this effect (*p* < 0.05) (Figure 3D,H,L,M,N), returning this parameter to values similar to the control group.

### 3.4. C4 Modifies the Inflammatory Environment

Ang-II leads to the chronic, low-grade synthesis of proinflammatory cytokines such as IL-1β, IL-6, and TNF-α in the vascular wall, which impairs endothelial function [18]. To probe the inflammatory environment, these cytokines were quantified via ELISA in the kidney, which is a target organ of ED. As shown in Figure 4, Ang-II increased (*p* < 0.05) the production of IL-1β (10,823.2 ± 1027 pg/mg; Figure 4A), TNF-α (4766.9 ± 269 pg/mg; Figure 4B), and TGF-β (4904.6 ± 471 pg/mg; Figure 4E), while keeping IL-6 and IL-10 levels unchanged (1090.8 ± 173 and 17,167.8 ± 2383 pg/mg; Figure 4C,D). Both losartan and C4 controlled the increase in IL-1β (9417.0 ± 546 and 8228.4 ± 874 pg/mg, respectively; Figure 4A), returning it to levels similar to those of the control group (7847.8 ± 847 pg/mg). However, both treatments failed to control the increase in TNF-α (Figure 4B). On the other hand, only C4 caused a significant increase in IL-10 levels (21,985.6 ± 2615.9 pg/mg, Figure 4E), whereas both C4 and Losartan reduced TGF-β concentration to control levels (3724.3 ± 214 and 3665.6 ± 214 pg/mg, respectively Figure 4E).

### 3.5. C4 Ameliorates Kidney Damage

To examine whether C4 reversed renal damage due to Ang-II-related ED, both renal function (as measured by creatinine serum levels) and the histopathological condition of the kidneys were analyzed. Mice were anesthetized at week 20, and peripheral blood and organs were obtained. Renal condition in the experimental groups is shown in Figure 5. As shown, Ang-II induced mononuclear cell infiltration in perirenal fat (Figure 5B), in the renal capsule (Figure 5F), and in the connective tissue of the microvasculature (Figure 5J). In these last two structures, thickened vascular walls due to collagen fibers are observed, related to the presence of fibroblasts. Finally, Ang-II induced glomerular hypertrophy (Figure 5N,Q; 706.2 ± 155.5 µm^2^), collagen deposits around glomeruli (Figure 5N), decreased glomerular capillary lumen (9.8 ± 2.0%; Figure 5S), and increased mesangial area (93.5 ± 9.0%; Figure 5R). Both Losartan and C4 decreased the amount of cellular infiltrate in perirenal fat and renal capsule (Figure 5C,D,G,H). C4 restored both structures to conditions similar to the control group (Figure 5A,E). On the other hand, both C4 and Losartan decreased renal capsule thickness (Figure 5H,G, respectively) as the amount of collagen fibers and fibroblasts was reduced. With respect to the microvasculature, both Losartan (Figure 5K) and C4 (Figure 5L) regenerated the vessels, bringing them back to conditions comparable to the control group (Figure 5I). Finally, C4 reduced glomerular hypertrophy (Figure 5P,Q; 628.1 ± 144.7 µm^2^) and mesangial expansion (88.0 ± 7.9%; Figure 5R); it also improved glomerular capillary (13.5 ± 2.3%; Figure 5S) to a similar extent as Losartan (610.9 ± 96.7 µm^2^; 88.3 ± 7.0% and 14.8 ± 2.5%, respectively).

To assess renal function in Ang-II-treated mice, serum creatinine was quantified. As shown in Figure 5T, Ang-II increased creatinine levels (1.15 ± 0.03 mg/dL; *p* < 0.05), which then were reduced to values similar to controls by both Losartan (0.92 ± 0.04 mg/dL) and C4 (0.81 ± 0.03 mg/dL), indicating that both treatments restored renal filtration.

### 3.6. C4 Ameliorates Liver Damage

ED, which is characterized by a low bioactivity of NO in the hepatic circulation (due to the combined effect of a reduced bioavailability of NO and its accelerated degradation by reactive oxygen species), has been reported to cause liver damage [19]. In a murine model of Ang-II-induced ED, we observed that not only the kidney but also the liver was damaged [11]. Since C4 prevented kidney damage, liver status was assessed by histopathological studies, organ weight, and triglyceride and glucose concentrations. As shown in Figure 6, mice from all groups developed fatty liver (Figure 6A–D). In Ang-II-treated animals, thickening of Glisson’s capsule (Figure 6F) and trabeculae (Figure 6J) due to the deposition of collagen fibers (10,660.4 ± 161.6 µm^2^, 11,647.4 ± 593.9 µm^2^; (Figure 6M,N) was observed (*p* < 0.05), as well as infiltration of fibroblasts, fibrocytes, and mononuclear cells such as lymphocytes and macrophages. Our results indicate the chronicity of the inflammatory state and the ongoing repair process. Treatment with C4 (Figure 6D,H,L,M,N) reversed these parameters, as did Losartan (Figure 6C,G,K,M,N), returning them to a state similar to the control group (Figure 6A,E,I). On the other hand, Ang-II induced hepatomegaly (1.8 ± 0.1 g), triglyceride accumulation (252.2 ± 31.7 mg/dL), and cellular infiltrate foci formation (6.6 ± 0.9 AU, Figure 6O–Q) (*p* < 0.05), in addition to lower glucose accumulation (168.1 ± 24.1 mf/dL; Figure 6R) (*p* < 0.05). Both Losartan and C4 prevented cellular infiltrate foci formation (2.0 ± 0.3 and 1.3 ± 0.2 AU; Figure 6O) and hepatomegaly (1.39 ± 0.15 and 1.44 ± 0.12 g, respectively, Figure 6P) (*p* < 0.05), but only C4 reversed triglyceride accumulation (196.7 ± 11.1 mg/dL, Figure 6Q) (*p* < 0.05); while C4 failed to normalize glucose concentrations (272.9 ± 17.5 mg/dL), these were significantly increased with respect to the animals treated with Ang-II only (168.1 ± 24.1 mg/dL) and Ang-II/Losartan (189.9 ± 26.5 mg/dL) (Figure 6R).

## 4. Discussion

Endothelial dysfunction is characterized by sustained vasoconstriction, VSMC hyperproliferation, oxidative stress, and a proinflammatory, prothrombotic status [5]. A major cause of ED is a decreased bioavailability of NO since this small molecule regulates various activities of the endothelium, including the control of vascular tone, platelet aggregation, endothelial permeability, and neoangiogenesis [7]. NO is produced in endothelial cells by eNOS. In turn, this enzyme is positively regulated by the serine/threonine kinase Akt; thus, this kinase is a key control point for eNOS activation and endothelial dysfunction [7,20] and a clear therapeutic target. In this work, it was shown that C4 treatment induced Akt activation and NO production in dysfunctional endothelial cells (Figure 2). This effect could be due to the presence in C4 of glycine, arginine, and aspartic acid, the main constituents of SF1 and SF3 [15]. Glycine has been reported to increase Akt phosphorylation while decreasing the expression of its negative regulator PTEN [21,22]. Glycine also increases the expression of eNOS mRNA and the enzyme itself [23,24]. On the other hand, arginine, in addition to being the substrate of eNOS, can stimulate the phosphatidylinositol-3-kinase (PI3K)/Akt pathway, favoring eNOS activation, NO production, and its vasodilator effects, thus improving endothelial function [25,26,27]. Finally, while aspartic acid is not directly involved in NO synthesis, this amino acid can be converted to arginosuccinate via arginosuccinate synthetase and then to arginine by arginosuccinate lyase; therefore, increased availability of the substrate for eNOS favors NO production [28]. By increasing the bioavailability of NO, C4 may offset several pathologic traits of ED. Although mice continuously received Ang-II stimulation, vascular remodeling was significantly improved in our study (Figure 3). In addition to the effects of glycine, arginine, and aspartic acid on Akt activation and NO availability, this finding could also be due to increased expression of the silent information regulatory protein 1 (SIRT1) homolog by arginine [29]. This protein acts as an activator of histone deacetylase protein-1 (AP-1), modulating neointimal layer formation and reducing the proliferation and migration of VSMCs [30]. As a result of the mitigation of vascular remodeling, BP returns to normal levels (Figure 1).

Another relevant marker of DE is inflammation. In this work, C4 promoted a significant increase in IL-10 expression and decreased that of IL-1β. IL-10 is an anti-inflammatory cytokine that suppresses the activation of the IL-1 converting enzyme and the subsequent processing and release of mature IL-1β [31]. This effect could be due to glycine and arginine, as glycine has been reported to increase IL-10 production [32], whereas arginine epigenetically regulates IL-10 via the hypomethylation of its gene promoter [33]. Furthermore, pro-IL-1β production is partially mediated by NF-κB activation [34]. Glycine and arginine have been reported to have anti-inflammatory effects by negatively regulating NF-κB activity, inhibiting the expression of IL-6, IL-1β, IL-17, TNF-α, and cyclooxygenase 2, as well as macrophage infiltration [23,26,35,36,37]. Despite these anti-inflammatory effects, neither Losartan nor C4 controlled the increase in TNF-α in this work. Another signaling pathway involved in the production of TNF-α is MAPK p38 [38]; thus, although glycine and arginine inhibited NF-kB activation, they may not inhibit the p38 pathway [38].

In addition to inflammation and oxidative stress, ED causes tissue damage in different organs, particularly the kidney and liver [39]. In renal arterioles, glomerular capillaries, and peritubular capillaries, ED causes impairment of renal function and contributes to the development of acute and chronic kidney diseases [40] such as glomerulosclerosis, characterized by an increase in mesangial volume [41,42]. Ang-II-induced ROS production leads to hyperplasia and hypertrophy of various cell types and increases the expression of cytokines and adhesion molecules [43]. These alterations promote the synthesis of extracellular matrix proteins by activating mesangial cells and interstitial and tubular fibroblasts, stimulating plasminogen activator inhibitor 1 (PAI-1), and promoting macrophage infiltration and activation, which further potentiates inflammation and fibrosis [15,43]. In this work, treating mice with Ang-II resulted in renal structural alterations such as the deposition of collagen fibers, infiltration of mononuclear cells (lymphocytes, macrophages, and fibroblasts), and an increase in mesangial and glomerular area (Figure 5). C4 reversed renal damage, possibly due to the anti-inflammatory effect described above, by inhibiting the activation of the involved cells and macrophages. Indeed, activation of the glycine receptor (GlyR, an ionotropic or ligand-gated receptor) has been reported to contribute to the hyperpolarization of macrophages, hindering their activation [44]. On the other hand, the control of fibrosis in various tissues could be due to the content of arginine. Ramprasath et al. [45] reported that arginine increased the expression of Nrf2, a transcription factor that promotes the expression of cytoprotective genes such as phase-2 enzymes (heme oxygenase-1, glutathione-S-transferase A1, and NAD(P)H quinone oxidoreductase 1) [46], which can offset the fibrinogenic and oxidative damage [47,48,49].

Using in vivo [11,50,51] and in vitro [52] models, several authors have shown that Ang-II causes hepatic steatosis, increasing liver weight and triglyceride concentrations. However, there are few reports on the effect of this hormone on lipid synthesis [53]. In this study, we found that Ang-II decreased the concentration of glucose in the liver, in line with the results of Wu et al. [54] on the effect of Ang-II on HepG2 hepatocytes. Those authors demonstrated that Ang-II activates the transcription factor SREBP2, involved in de novo lipogenesis, a process that uses glucose as a raw material to produce oleic and palmitic acid [53], which are immediately esterified into triglycerides [54]. Previously, we observed that chronic application of Ang-II in mice induces steatosis by activating the de novo lipogenesis pathway [51], an effect that would be evidenced by lower intracellular glucose concentrations [51,54]. We also observed that Losartan increased triglyceride concentrations and decreased glucose levels within the hepatocyte, in agreement with the report by Schupp et al. [55] that Losartan induced steatosis via activation of SREBP2 and PPAR. Meanwhile, C4 reduced triglyceride levels and partly restored glucose concentration, possibly due to the glycine and arginine content in C4, as these amino acids have been reported to have anti-steatosis activity in vivo [56,57].

Steatosis itself induces lipid degradation, which in turn leads to oxidative stress and, ultimately, cell death due to necrosis and inflammation, which together promote collagen fiber deposition [58]. The binding of Ang-II to the AT1 receptor of Kuppfer cells and hepatocytes increases the production of proinflammatory cytokines, including chemokines such as CXCL8 [59,60], triggering an immune response by attracting more cells to the site of injury. In addition, Ang-II has been reported to activate hepatic stellate cells via ATR1, promoting the development of fibrosis by increasing the expression of collagen I by induction of the intracellular MAPK/ERK pathway. It also promotes an increase in tissue inhibitor of metalloproteases-1 (TIMP-1) in the liver, producing disequilibrium with increased collagen and protein deposition in the extracellular matrix (ECM) [61,62]. Interestingly, C4 decreased both cellular infiltrate and fibrosis in mice, possibly due to its content of glycine, which has been reported to inhibit inflammation and liver fibrosis; however, its mechanism of action is still unknown [36,63,64].

## 5. Conclusions

Overall, we found that C4 promotes the Akt/eNOS pathway and improves NO production and endothelial functions, helping to regulate BP, vascular remodeling, and inflammation whilst ameliorating tissue damage. These effects support its potential use as a phytomedicine.

## Figures and Tables

**Figure 1 nutrients-15-04680-f001:**
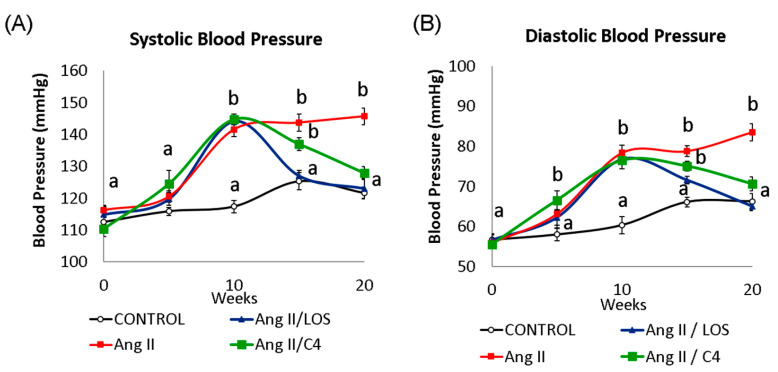
Blood pressure kinetics: (**A**) systolic (SBP) and (**B**) diastolic (DBP). Both blood pressure values were measured in all mice every 5 weeks throughout the experiment (20 weeks). At week 10, all Ang-II-treated mice showed a significant increase in blood pressure from baseline (week 0). Administration of either C4 or Losartan started at week 10. Losartan and C4 decreased both pressure values at week 5 of treatment, with Losartan being more effective. Data are reported as mean ± SE, and they were analyzed by ANOVA and Tukey’s post hoc test. Different letters indicate significant differences (*p* < 0.05) between groups (*n* = 15 animals per group).

**Figure 2 nutrients-15-04680-f002:**
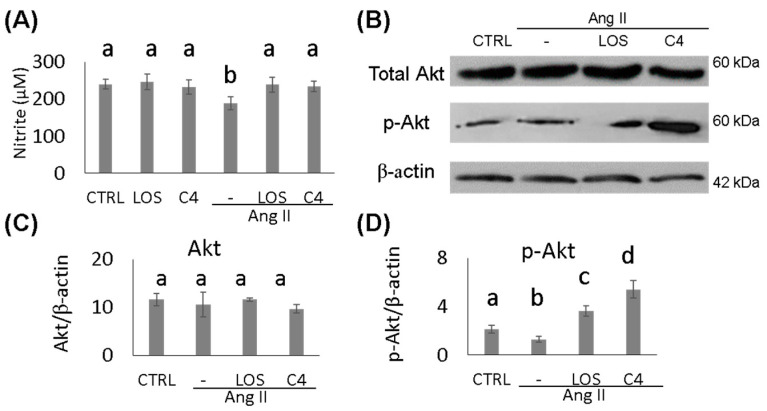
Nitrite quantification and expression of AKT and p-AKT. (**A**) Nitrite quantification (*n* = 7). (**B**) Western blot of AKT and phospho-AKT (*n* = 2). (**C**,**D**) AKT and phospho-AKT expression. Ang-II treatment reduced NO availability, while both treatments prevented its decrease, which can be linked to AKT phosphorylation. Data are reported as mean ± SE, and they were analyzed by ANOVA and Tukey’s post hoc test. Different letters indicate significant differences (*p* < 0.05) between groups.

**Figure 3 nutrients-15-04680-f003:**
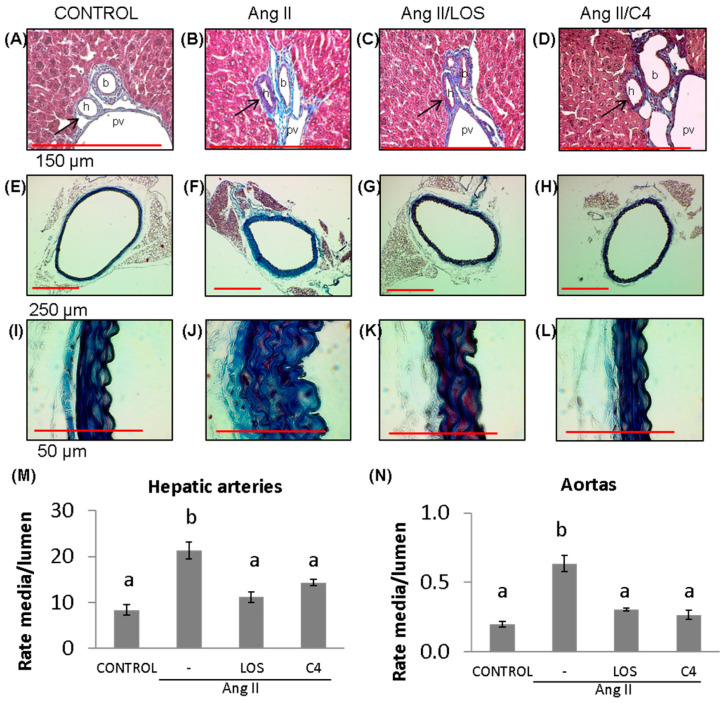
Vascular remodeling. (**A**–**D**) Photomicrographs of hepatic arteries, 40× (red bar = 150 μm). (**E**–**L**) Photomicrographs of aorta stained with Masson’s trichrome (4×, red bar = 250 μm, (**E**–**H)**; 100×, red bar = 50 μm, (**I**–**L**)). (**M**) Microvascular remodeling, determined as the media/lumen ratio in hepatic arteries (*n* = 10 hepatic arteries, two for each mouse). (**N**) Macrovascular remodeling, determined as the media/lumen ratio in aorta (*n* = 5 aorta). Ang-II induced remodeling in micro- (hepatic arteries) and macrovasculature (aorta); both Losartan and C4 reversed this effect. Data are reported as mean ± SE, and they were analyzed by ANOVA and Tukey’s post hoc test. Different letters indicate significant differences (*p* < 0.05) between groups. LOS: Losartan; C4: combination of subfractions SF1 and SF3; b: bile duct; pv: portal vein; h: hepatic artery. Arrows show the hepatic artery.

**Figure 4 nutrients-15-04680-f004:**
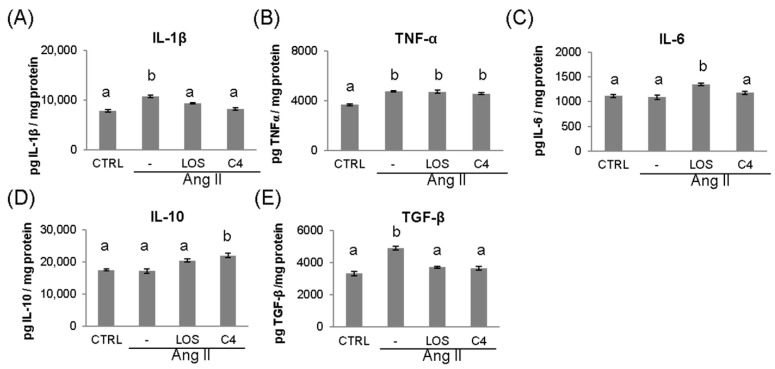
Concentration of pro- and anti-inflammatory cytokines in kidney. The levels of proinflammatory (IL-1β, TNF-α, and IL-6) (**A**–**C**) and anti-inflammatory cytokines (TGF-β and IL-10) (**D**,**E**) were measured in the kidney at the end of the experiment (week 20). Data are reported as mean ± SE, and they were analyzed by ANOVA and Tukey’s post hoc test. Different letters indicate significant differences (*p* < 0.05) between groups. LOS: Losartan; C4: combination of subfractions SF1 and SF3. *n* = 10 mice per group.

**Figure 5 nutrients-15-04680-f005:**
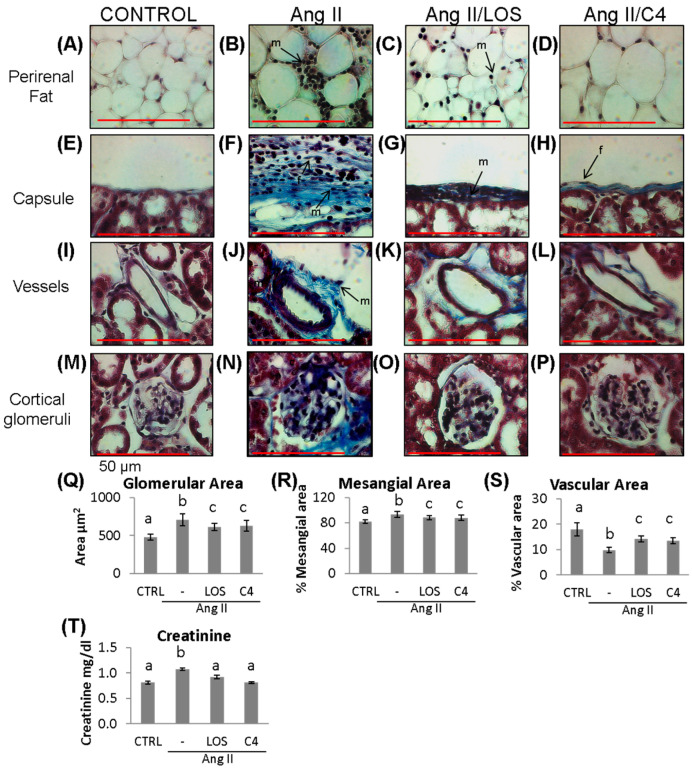
Kidney damage. Representative images of (**A**–**D**) perirenal fat; (**E**–**H**) renal capsule; (**I**–**L**) vessels; and (**M**–**P**) cortical glomerulus of control and Ang-II-, Ang-II/Losartan-, and Ang-II/C4-treated mice. Analysis of glomerular damage (**Q**–**S**) and renal profile (**T**) (*n* = 10). Ang-II induces inflammation, evidenced by infiltrating mononuclear cells and fibroblasts and by deposition of collagen fibers, which were stained in blue (**B**,**F**,**J**,**N**). Treatment with C4 decreases these parameters (**D**,**H**,**L**,**P**) better than Losartan (**C**,**G**,**K**,**O**). Ang-II induced glomerular hypertrophy (**Q**) and mesangial expansion (**R**) while decreasing glomerular capillarity (**S**). Treatment with C4 partially reversed these parameters. Ang-II-induced increased serum creatinine levels, indicative of a compromised kidney function, were reverted by C4 and Losartan (**T**). Photomicrographs were taken at 100× (red bar = 50 μm). Data are reported as mean ± SE, and they were analyzed by ANOVA and Tukey’s post hoc test. Different letters indicate significant differences (*p* < 0.05) between groups. LOS: Losartan; C4: combination of subfractions SF1 and SF3; m: mononuclear cell; f: fibroblasts.

**Figure 6 nutrients-15-04680-f006:**
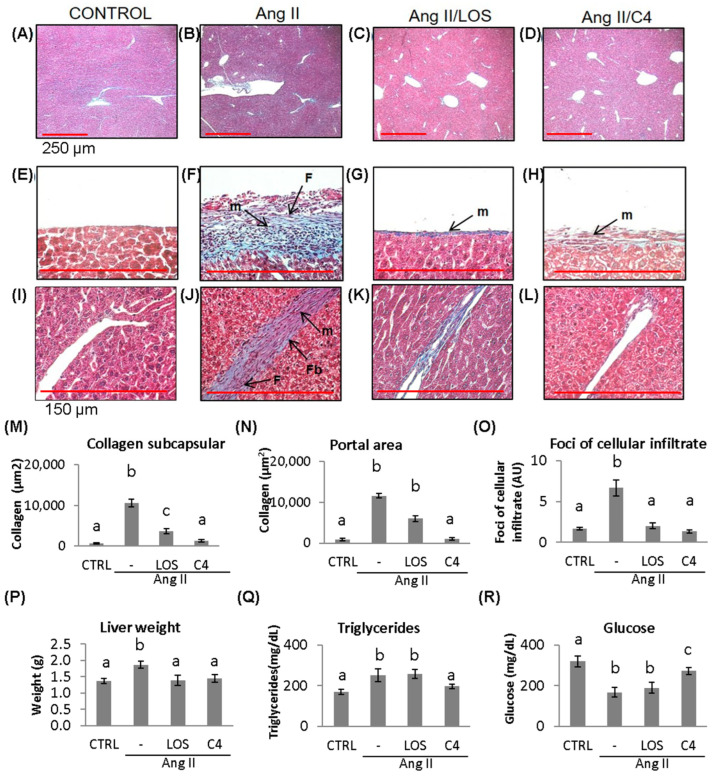
Liver damage assessment by Masson’s trichrome stain. Representative images of (**A**–**D**) hepatic parenchyma, 4×, (red bar = 250 μm). (**E**–**H**) Glisson’s capsule, 40×, (red bar = 150 μm); (**I**–**L**) trabeculae, 40×, control, Ang-II-, Ang-II/Losartan-, and Ang-II/C4-treated mice. (**M**,**N**) Measurement of collagen fiber deposition and (**O**) cellular infiltrate foci formation. Liver state assed assessed by liver weight and triglycerides and glucose concentration (**P**–**R**). Ang-II-treated mice showed steatosis, mononuclear cell infiltrate, and collagen fiber deposition. Both Losartan and C4 decreased Glisson’s capsule and trabeculae thickening. Photomicrographs taken at 4× and 40×. Data are reported as mean ± SE, and they were analyzed by ANOVA and Tukey’s post hoc test. Different letters indicate significant differences (*p* < 0.05) between groups (*n* = 5). LOS: Losartan; C4: combination of subfractions SF1 and SF3; m: inflammatory mononuclear cells; F: fibroblasts; Fb: fibrocytes.

## Data Availability

Not applicable.

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
