# Peer review of "Aqueous Fraction from Cucumis sativus Aerial Parts Attenuates Angiotensin II-Induced Endothelial Dysfunction In Vivo by Activating Akt"

_nutrients, 2023, doi:10.3390/nu15214680_

Round 1
Reviewer 1 Report
Comments and Suggestions for Authors
The manuscript "Aqueous fraction from Cucumis sativus aerial parts attenuates angiotensin II-induced endothelial dysfunction in vivo by activating Akt" indicates the protective and medicinal role of the aqueous fraction from cucumber. The research was carried out at a very high level, the experiments are evidence-based, and the results obtained are very important in terms of searching for new antihypertensive drugs. The only thing that would be very interesting is to know which component has such a pronounced protective effect on the vascular endothelium. And it would be very interesting to study mass spectrometry of compounds in the aqueous fraction of cucumber.
Author Response
The manuscript "Aqueous fraction from Cucumis sativus aerial parts attenuates angiotensin II-induced endothelial dysfunction in vivo by activating Akt" indicates the protective and medicinal role of the aqueous fraction from cucumber. The research was carried out at a very high level, the experiments are evidence-based, and the results obtained are very important in terms of searching for new antihypertensive drugs. The only thing that would be very interesting is to know which component has such a pronounced protective effect on the vascular endothelium. And it would be very interesting to study mass spectrometry of compounds in the aqueous fraction of cucumber.
R:
Thank you for your kind and timely review or our study.
With respect to the molecules involved in vascular endothelial protection, we had previously reported that the major components in the SF1 and SF3 subfractions constituting C4 are the amino acids glycine, arginine, asparagine, lysine, and aspartic acid (reference 15), and mass spectrometry results supporting this finding were provided as supplemental figures in that manuscript. Of these amino acids, glycine has some reported effects on endothelial cells or other cell types involved in the development of endothelial dysfunction. One such effect is its ability to induce Akt phosphorylation and decrease the expression of PTEN, which is the negative regulator of Akt; it also increases the expression of both eNOS mRNA and the protein itself (lines 377–379); another effect is an increase in the production of the anti-inflammatory cytokine IL-10 (line 399) and a decrease in the activation of NF-κB and, therefore, in the production of some proinflammatory mediators (lines 401–404). In addition, glycine decreases macrophage activation, another cell type known to actively participate in endothelial dysfunction (lines 421–425). All these effects, previously reported for glycine, are reviewed in the Discussion section of our manuscript. Thus, this amino acid, present in the extract, could be exerting important protective effects on endothelial cells.
On the other hand, in addition to being the substrate of eNOS, arginine has effects on the PI3K/Akt/eNOS signaling pathway that favor endothelial cell function (lines 380–382). It also increases the expression of SIRT1, a protein with anti-remodeling effects in VSMCs (lines 390–392); arginine epigenetically regulates IL-10 production (lines 399–400), increases Nrf-2 (which favors the expression of cytoprotective genes) (lines 427–430) and decreases NF-κB activation (lines 402–404).
Another amino acid in C4 is aspartic acid. Although no previous work has reported that it acts directly on the endothelium, it may serve as a substrate for arginine production, which in turn is the substrate for NO production by eNOS (lines 382–385), so it could also be promoting the maintenance of a healthy endothelium.
Taken altogether, the activity of the amino acids found in C4 protects and reverses the deleterious effects of endothelial dysfunction. While it would be interesting to evaluate the effect of each amino acid in C4, this will be addressed in a future study.

Reviewer 2 Report
Comments and Suggestions for Authors
This is a very interesting work focus on how to treat endothelial dyfunction, thus I would like to recommend this manuscript for publication after minor revision:
1. The language spelling should be carefully checked although it does not influence reading, such as H2SO4 in Line 184, not rigorous.
2. In Figure 2B, I suggest the authors to repeat this experiment because in the first line Strands entangled together can affect the accuracy of data, and there are remnants in the third row.
3. Immunohistochemical images in Figure 3, 5 and 6 should be added with scale bar.
4. There are many factors that affect endothelial cell function, such as macrophage phenotype, smooth muscle cell phenotype, etc. The authors can discuss them in combined with suggested literature or briefly describe them in the Introduction section <Comparison of conjugating chondroitin sulfate A and B on amine-rich surface: for deeper understanding on directing cardiovascular cells fate, Composites Part B: Engineering 228 (2022) 109430.>.
Author Response
Thank you for your kind and timely review or our study.
- The language spelling should be carefully checked although it does not influence reading, such as H2SO4 in Line 184, not rigorous.
R: Thank you for pointing this. The manuscript was revised again by an expert English editor, and appropriate changes in spelling were made.
- In Figure 2B, I suggest the authors to repeat this experiment because in the first line Strands entangled together can affect the accuracy of data, and there are remnants in the third row.
R: The blots in the original manuscript were replaced with images of those same blots taken 10 seconds earlier, to prevent band saturation (figure added into the revised manuscript). Also, we provided the repetition of this assay during submission, which were not added in the PDF.
- Immunohistochemical images in Figure 3, 5 and 6 should be added with scale bar.
A: We apologize for this omission. Scale bars were added to Figures 3, 5, and 6, with corresponding captions. As you will note, adding the scale bars to each image in the figure makes them too cluttered and difficult to understand. In our opinion, it would be better to add a scale bar to the first image of each experimental group and indicate in the caption that all other images are to scale with the corresponding ones.
- There are many factors that affect endothelial cell function, such as macrophage phenotype, smooth muscle cell phenotype, etc. The authors can discuss them in combined with suggested literature or briefly describe them in the Introduction section <Comparison of conjugating chondroitin sulfate A and B on amine-rich surface: for deeper understanding on directing cardiovascular cells fate, Composites Part B: Engineering 228 (2022) 109430.>.
A: We agree with the reviewer. Indeed, endothelial cell function can be affected by several factors. Thank you for helping us improve our manuscript by adding this information and for providing an appropriate reference. A brief paragraph to this effect was added on lines 43–45, in the Introduction.

Reviewer 3 Report
Comments and Suggestions for Authors
Good work that experimentally substantiates the feasibility of phytotherapy. There are some uncertainties that should be cleared up: 1) what is the rationale for the selection of animals (exactly 5 animals) in each group?
2) in some places it is difficult to understand the description of results. For example, Figure 1 mentions (row 229) "Different letters indicate significant differences (p < 0.05) between groups (n = 15)", but 4 groups were compared, so 20 animals.
3) When describing the dynamics of indicators, it would be good to show the dynamics in absolute numbers at different stages of the study, not just the dynamics in percentages (rows 212-230)
Author Response
Good work that experimentally substantiates the feasibility of phytotherapy.
Thank you for your kind and timely review or our study.
There are some uncertainties that should be cleared up: 1) what is the rationale for the selection of animals (exactly 5 animals) in each group?
R: For Figure 1, 5 animals per group were included, but the experiment was repeated three times to make sufficient tissue samples available for ELISA and histological studies. Based on previous studies on the extract and following ethical guidelines to minimize the number of animals used in the experiments, 5 animals were included in each group. This still provides adequate statistical power. As the reviewer will see in the response below, out of a total of 15 animals per group, tissue samples from 10 specimens were used for ELISA assays and the other five samples for histological studies.
2) in some places it is difficult to understand the description of results. For example, Figure 1 mentions (row 229) "Different letters indicate significant differences (p < 0.05) between groups (n = 15)", but 4 groups were compared, so 20 animals.
A: We apologize for this confusion. As mentioned above, data in Figure 1 correspond to 15 animals for each experimental condition. Five animals were used for each independent trial. Thus, a total of 60 animals were included in the four experimental groups. Of these 15 animals per group, tissues from 10 mice were used for ELISA assays and the other five for histological studies.
3) When describing the dynamics of indicators, it would be good to show the dynamics in absolute numbers at different stages of the study, not just the dynamics in percentages (rows 212-230)
R: To address the reviewer’s suggestion, absolute numbers are provided in the revised manuscript.
